# Soft Magnetic Amorphous Microwires for Stress and Temperature Sensory Applications

**DOI:** 10.3390/s19235089

**Published:** 2019-11-21

**Authors:** Larissa Panina, Abdukarim Dzhumazoda, Makhsudsho Nematov, Junaid Alam, Alex Trukhanov, Nikolay Yudanov, Alexander Morchenko, Valeria Rodionova, Arcady Zhukov

**Affiliations:** 1Institute of Novel Materials and Nanotechnology, National University of Science and Technology (MISiS), Moscow 119991, Russia; Abdukarim_jumaev@mail.ru (A.D.); nmg_1988@mail.ru (M.N.); engr.sajalam@gmail.com (J.A.); truhanov86@mail.ru (A.T.); kolyan2606@mail.ru (N.Y.); dratm@mail.ru (A.M.); valeriarodionova@gmail.com (V.R.); 2Institute of Physics, Mathematics & IT, Immanuel Kant Baltic Federal University, Kaliningrad 236041, Russia; 3Institute for Design Problems in Microelectronics RAS, Moscow 124681, Russia; 4Department Materials Physics, University Basque Country, UPV/EHU, 20018 San Sebastian, Spain; arkadi.joukov@ehu.eus; 5Ikerbasque, Basque Foundation for Science, 48013 Bilbao, Spain

**Keywords:** amorphous microwire, low Curie temperature, annealing, current annealing, near-zero magnetostriction, induced anisotropy, harmonics spectrum, magnetic bistability, magnetoimpedance

## Abstract

Amorphous ferromagnetic materials in the form of microwires are of interest for the development of various sensors. This paper analyzes and argues for the use of microwires of two basic compositions of Co_71_Fe_5_B_11_Si_10_Cr_3_ and Fe_3.9(4.9)_Co_64.82_B_10.2_Si_12_Cr_9(8)_Mo_0.08_ as stress/strain and temperature sensors, respectively. The following properties make them suitable for innovative applications: miniature dimensions, small coercivity, low anisotropy and magnetostriction, tunable magnetic structure, magnetic anisotropy, and Curie temperature by annealing. For example, these sensors can be used for testing the internal stress/strain condition of polymer composite materials and controlling the temperature of hypothermia treatments. The sensing operation is based on the two fundamental effects: the generation of higher frequency harmonics of the voltage pulse induced during remagnetization in wires demonstrating magnetic bistability, and magnetoimpedance.

## 1. Introduction

Amorphous alloys based on 3D metals produced in the form of ribbons and wires are excellent soft magnetic materials used in a large number of applications, including sensors [1,2,3,4]. The choice of composition depends on a particular application, but generally, the alloy includes (Fe,Co,Ni)_70–85_(Si,B)_15–30_ where the metalloids Si and B help glass formation [5]. The addition of other elements is also used, such as, for example, Cr and Mo to stabilize the amorphous structure [6,7,8,9]. Here, we consider microwires made of a number of alloy systems that have high potential for the development of miniature mechanical stress and temperature sensors based on their tunable magnetic properties. The wires are prepared by the modified Taylor-Ulitovskiy method [10,11], and when quenched in water, they typically have an amorphous structure without crystalline phases. Then, the residual magnetocrystalline anisotropy is small, and the magnetoelastic interactions often play the deciding role in determining the overall magnetic anisotropy energy. This is the basis of magnetic stress-sensing mechanisms with the use of amorphous materials [12,13,14,15,16,17,18].

For many sensing applications, temperature stability is of primary importance, which demands the use of materials with a high Curie temperature, $T_{c}$. However, for ambient temperature measurements, low-$T_{c}$ materials are sought after. Microwires made of amorphous CoFe-based alloys with a higher addition of Cr or Ni have reduced values of $T_{c}$ [6,14,19,20,21] and are suitable for local temperate measurements.

In amorphous materials, the short-range ordering can be modified by various annealing treatments [22,23]. This is important from the perspective of tuning the magnetic structure and controlling the magnetic parameters such as uniaxial anisotropy, magnetostriction, and Curie temperature [24,25,26,27]. In particular, annealing in the presence of a magnetic field forms the easy anisotropy axis along the field if the annealing temperature is lower than the Curie temperature. Simultaneously, the internal stress frozen-in during fabrication is relaxed, which eliminates the temperature instability. For example, the current annealing of amorphous wires of Co-rich compositions with moderate current intensities induces a circumferential easy anisotropy and positive magnetostriction [24,28,29]. The application of tensile stress along the wire contributes to long-range axial anisotropy and strongly affects the remagnetization process and magnetoimpedance. Therefore, this magnetic configuration is desirable for stress/strain sensing with the use of microwires and is in the focus of this work.

The current annealing of amorphous microwires in different modifications was intensively used with the aim to establish a well-defined circumferential anisotropy [30,31]. This anisotropy is the key to realize large and sensitive change in the complex-valued impedance in the presence of a DC magnetic field known as the magnetoimpedance effect (MI) [32,33,34]. The relative change in impedance (the MI ratio) in the range of 400%–600% was obtained. Since current annealing relaxes the internal stresses, excellent temperature stability of the magnetic structure and MI was demonstrated. However, for mechanical sensing applications, a balance between the induced anisotropy and modifications in magnetostriction is crucially important. The desirable combination requires the optimization of the alloy composition and annealing conditions. Thus, the required properties are achieved in microwires of near-zero magnetostriction alloy of composition Co_71_Fe_5_B_11_Si_10_Cr_3_ with the metal core diameter of about 25 microns by annealing with the current densities of 100–120 A/mm^2^.

All characteristic magnetic parameters including the saturation magnetization, anisotropy, and magnetostriction decrease when the temperature approaches $T_{c}$ [35,36,37,38]. Specific sensing applications require good high-temperature performance. As the Curie temperature of FeCo-based alloys is typically higher than 300 ℃, these materials are suitable for such applications. On the other hand, considering possible applications in miniature temperature sensors operating in the industrial temperature range (room temperature–100 ℃), microwires with a reduced and tunable Curie temperature are needed [39]. CoFe-based amorphous alloys containing Cr have a reduced $T_{c}$, owing to the antiferromagnetic coupling between Cr-Fe and Cr-Co atoms. With increasing Cr concentration, the value of $T_{c}$ initially drops by about 24–25 ℃ per at% of Cr, and an even stronger decrease takes place if the Cr concentration is above 10% [6]. Here, we discuss the behavior of critical magnetic parameters near the Curie temperature in amorphous microwires of composition Fe_3.9_Co_64.82_B_10.2_Si_12_Cr_9_Mo_0.08_ with a low $T_{C}$ of 62 ℃. The wires have an axial anisotropy due to a positive magnetostriction. The easy axial anisotropy is important to realize fast remagnetization even near $T_{c}$. It is also favorable for temperature-dependent MI, although rather high frequencies of a few hundred megahertz are needed to observe a monotonic drop in the impedance with increasing temperature toward $T_{c}$.

The compositional adjustment of $T_{c}$ within a required narrow range is difficult, but this may be achieved by annealing causing the microscopic atomic rearrangements [23,40,41]. Typically, $T_{c}$ variations are less than 10% when temperatures are defined in Kelvin. However, in absolute values, this can be within 10–15 degrees, which is sufficient for certain requirements on the temperature range. In the case of biomedical applications, the Curie temperature of 40–60 ℃ would be of interest.

The paper is organized as follows. Section 2 gives the details of the experimental and technological procedures used in this work. Section 3 provides a theoretical background of stress/temperature tunable static and dynamic magnetic properties. Section 4.1 analyzes the impact of mechanical stress on the magnetization processes in glass-coated amorphous microwires of composition Co_71_Fe_5_B_11_Si_10_Cr_3_ before and after current annealing. In Section 4.2, the temperature-dependent magnetic properties of amorphous microwires of composition Fe_3.9(4.9)_Co_64.82_B_10.2_Si_12_Cr_9(8)_Mo_0.08_ with low $T_{c}$ values of 61–62 ℃ are considered. Section 4.3 introduces the sensing mechanism based on the fast remagnetization and generation of a high harmonics spectrum. In Section 4.4, we discuss the impact of mechanical stress on MI and the role of a particular magnetic configuration to realize the stress-MI effect. Here, we also consider the MI change when approaching the Curie temperature. The main findings of the paper are summarized in the Conclusion.

## 2. Materials and Methods

Microwires prepared by the Taylor-Ulitovisky method [10] in a glass coating are discussed in this work. For stress-sensory applications, microwires with a low magnetostriction constant are of interest. This is realized in Co_x_Fe_y_ and Co_x_Mn_y_ alloys with a high content of Co (x/y≈14) [42,43]. Alloys with Ni also have a low magnetostriction, but the saturation magnetization and the Curie temperature are also reduced [44]. Considering this reasoning, glass-coated amorphous wires of composition Co_71_Fe_5_B_11_Si_10_Cr_3_ (x/y≈14.2) were chosen to realize stress-sensitive magnetic structures. A small addition of Cr is helpful to stabilize the amorphous state [45]. The Curie temperature $T_{c}$ is 637 K. The wire dimensions were a total diameter of D≈30 µm and a metal core diameter of d≈ 25 µm.

Reduced values of $T_{c}$ are achieved in alloys with a high content of Cr or Ni. To demonstrate the temperature effects near $T_{c}$, the microwires of the Fe_3.9(4.9)_Co_64.82_B_10.2_Si_12_Cr_9(8)_Mo_0.08_ composition with the Curie temperature in the range of 334.5−345 K were chosen. The wire had a total diameter of D≈27 µm and a metallic core diameter of d≈ 17.5 µm.

For property modification, the wires were annealed using furnace heating and Joule heating by DC current. The latter is of interest for the formation of the induced anisotropy by a circular magnetic field of the current. The sample length for current annealing was 15 cm, the passed current amplitude varied in the range of 25–90 mA, and the flowing current time varied within up to 60 minutes. All the current treatments were performed in the same ambient conditions. During current annealing, the temperature control is of high importance, since the annealing temperature $T_{an}$ should be sufficiently high to ensure fast kinetics but not exceed the Curie temperature. One method to measure $T_{an}$ is to make a comparison between the dependences of the saturation magnetization on the temperature and annealing current [23]. The value of $T_{an}$ can be also found from modeling, considering the balance between the supplied electric power and heat exchange [46]. We have recently proposed a direct method of measuring the temperature during current annealing, which is based on using a reference sample, the temperature of which is automatically set equal to the temperature of the annealed microwire. The setup uses a precision operational amplifier with a bridge switching circuit and a heating platform, as demonstrated in Figure 1.

The Curie and crystallization temperatures were determined from differential scanning calorimetry (DSC) curves (by using DSC 204 F1 Netzsch instrumentation) with the help of standard IT application. The crystallization temperature for the chosen alloys is in the range of $T_{cr}=773-793\;K$. For fast analysis, the Curie temperature of the annealed samples was determined from the temperature behavior of the AC magnetic susceptibility measured by an RLC meter at a frequency of 1 kHz. The samples were placed in a thermally isolated camera, and the temperature at the sample position was measured by a thermocouple.

The hysteresis loops under the effect of applied stress of up to 1.2 GPa and temperature (up to 80 ℃) were obtained by using the induction method with two differential detection/magnetization coils with an inner diameter of 3 mm each. The magnetization coil produced a magnetizing field with an amplitude of up to 1000 A/m and was driven by a current with a frequency of 500 Hz. The field magnitude was sufficient for remagnetizing the soft magnetic wires at this frequency. The external stress was applied by hanging the load at one end of the microwires, whilst the other was kept fixed. For elevated temperature measurements, the coil with the wire sample was placed in the thermal camera.

The Hewlett-Packard 8753E Vector Network Analyzer was used to measure the wire impedance in the frequency range between 1 and 500 MHz at room temperature. The wire length for impedance measurements was 11 mm. The impedance spectra were deduced from the S_21_ parameter (two-port measuring scheme) after making the calibration procedure with specially designed PCB-microstrip cells [47]. The sample was placed inside a Helmholtz coil that produced a slowly varying magnetic field up to 3000 A/m. In order to introduce a tensile stress during the S_21_ measurement, the load was hung by a thread in the middle part of the wire. In this case, the stress is not uniform along the wire length, and the estimated average value was used for the quantization of its effect. For measuring the impedance versus temperature, the change in temperature (from room temperature up to 370 K) was made with a hot-air gun and measured by a standard thermocouple.

A high harmonics spectrum was calculated through a digital sampling of the induced voltage signal measured during the wire remagnetization in the time domain and converting it into the frequency domain using fast Fourier transforms. Experimentally, the amplitudes of the chosen high harmonics (5th–15th) were measured with the help of a lock-in amplifier (lock-in nanovoltmeter type-232B) and functional generators. In order to measure the effect of temperature on the harmonics amplitude, a specially designed flat coil and vacuum chamber were used.

Considering the effect of applied load exerting the force P to the whole wire with a layered structure consisting of a metal core and a surface glass shell, the value of stress ${\hat{\sigma }}_{ex}$ in the core needs to be evaluated. The deformations in the inner metal core and the glass shell are assumed to be independent; then, the Poisson ratios are the same. In this case, the force P imposes the following tensile stresses in the core ($\sigma_{ex,\;in})$ and shell regions ($\sigma_{ex,\;out})$ [48]:(1)$$\sigma_{ex,\;in}=E_{m}C_{0},\;\sigma_{ex,\;out}=E_{g}C_{0}.$$
In Equation (4), $E_{m}$ and $E_{g}$ are the Young modules, while the constant $C_{0}$ is found from the definition of the average stress ${\bar{\sigma }}_{ex}$:(2)$${\bar{\sigma }}_{ex}=\frac{4P}{\pi D^{2}}=\left(E_{m}\rho^{2}+E_{g}\left(1-\rho^{2}\right)\right)C_{0},\;\;\rho =\frac{d}{D}.$$

In amorphous alloys, the values of the Young modules depend on the microstructure [49] and differ considerably for wires of similar composition. For example, the $E_{m}$ values of Co-rich amorphous wires range between 130 and 150 GPa, and the $E_{g}$ values for glass are within 50−70 GPa. Stress estimation in the metal core from Equations (4)–(5) was done with the use of the average values.

## 3. Theoretical Background

The soft magnetic properties of materials are largely required for applications in sensors based on their magnetization or remagnetization in weak magnetic fields. From this perspective, amorphous alloys are of great importance. The energy of magnetocrystalline anisotropy $E_{cr}$ in amorphous ferromagnetics is low, and the other contributions to the total anisotropy energy $E_{m}$, such as magnetoelastic $E_{me}$ and magnetodipole $E_{md}$ energies, can be also made small. In general, all these terms may be needed for analyzing the magnetic structure. Additionally, the induced anisotropy $E_{u}$ is established by various annealing treatments to refine the magnetic structure. The total energy $E_{m}$ is of the form:(3)$$E_{m}=\;E_{cr}+E_{u}+E_{me}+E_{md}$$
(4)$$E_{cr}=-\;K{\left(n_{k}\cdot m\right)}^{2},E_{me}=-\frac{3}{2}\lambda_{s}\left(\hat{\sigma }m\right)\cdot m,E_{u}=-K_{u}{\left(n_{u}\cdot m\right)}^{2}.$$

Here, $K,\;K_{u}$ and $n_{k,\;}n_{u}$ are the magnitudes and directions of the averaged crystalline anisotropy and induced anisotropy, respectively; $m=M/M_{s}$ is the unit magnetization vector, $M_{s}$ is the saturation magnetization, $\lambda_{s}$ is the saturation magnetostriction, and $\hat{\sigma }$ is the stress tensor, which is composed of internal stress ${\hat{\sigma }}_{in}$ occurring during material processing and the applied load ${\hat{\sigma }}_{ex}$. If the parameter $\lambda_{s}$ is small in the range of $10^{-8}-10^{-7}$, it is important to consider its stress dependence. A linear dependence between $\lambda_{s}$ and a tensile stress $\sigma_{a}$ was experimentally established [27,43,47,48,50]:(5)$$\lambda_{s}\left(\sigma_{a}\right)=\lambda_{s0}-\beta \sigma_{a}.$$

In Equation (3), $\lambda_{s0}$ is the magnetostriction in the stress-free state, and the parameter β is positive and rather small, residing in the range of $\;\left(1-6\right)\times 10^{-10}\;$/MPa. However, if $\lambda_{s0}$ is also small, the second term in Equation (3) may be critically important: with increasing $\sigma_{a}$, the negative contribution to $\lambda_{s}$ increases, which may result in a sharp change in the direction of the easy anisotropy. The possibility of transforming the type of magnetic anisotropy through changing the easy axis direction constitutes the basis of the development of highly sensitive sensing elements with respect to mechanical stresses.

### 3.1. Stress-Sensitive Magnetic Configurations in Amorphous Microwires

The following magnetic configurations in amorphous microwires may be exploited for stress-sensory applications:In many cases, the use of a DC bias magnetic field may create stress-sensitive magnetization reversal. This was exploited in wires with a negative magnetostriction and almost a circumferential easy anisotropy [3]. The application of an axial magnetic field deviates the magnetization toward the axis. The applied tensile stress strengthens the circular anisotropy and rotates the magnetization back. However, for practical applications, the use of a bias field may not be desirable.The as-produced wires of Co-rich composition may have an axial easy anisotropy and small magnetostriction (either positive or negative). In this case, the main contribution to the axial anisotropy can be due to Co–nanocrystal clustering in an amorphous matrix that has a high magnetic anisotropy, $K=\left(4.2-5.5\right)\times 10^{5}J/m^{3}$ for HCP Co [51]. The application of the external stress makes the magnetostriction larger in magnitude and negative according to Equation (5). The contribution of the magnetoelastic anisotropy increases and changes the direction of the anisotropy easy axis: it aligns closer to the circumference due to coupling between the negative magnetostriction and axial stress $\sigma_{a}$. The change in the anisotropy easy angle with increasing $\sigma_{a}$ for this case is shown in Figure 2. If the wire has a strong internal stress, then a circumferential anisotropy is observed without the applied stress, and thermal annealing forms the axial anisotropy due to relaxation of the frozen-in stress.Annealing in the presence of a magnetic field forms in wires a certain combination of anisotropy and magnetostriction due to atomic pair ordering, internal stress relaxation, and atomic rearrangements [52,53]. The microscopic origins of the induced anisotropy and transformation in magnetostriction are different, and their evolution during annealing may not correlate [24,27]. Recently, researchers demonstrated the possibility of inducing a circumferential anisotropy in combination with a positive magnetostriction in Co-rich amorphous wires [16]. Depending on the magnitude of the magnetostriction constant, the applied tensile stress may cause non-monotonic change in the easy anisotropy angle, as shown in Figure 3 (Curve 1). For higher values of magnetostriction, the application of tensile stress establishes the axial anisotropy (Figure 3, Curve 2).

### 3.2. Temperature-Sensitive Magnetic Configurations in Amorphous Microwires

Many FeCo-rich amorphous alloys have a high Curie temperature $T_{c}$ above 300 ℃ [54], but the magnetic structure of microwires produced from these alloys often experiences temperature variations at much lower temperatures below 80 ℃ [38,46,55,56,57,58,59]. This temperature influence is related with the internal stress relaxing and the corresponding transformation in the magnetoelastic anisotropy. Upon heating, a circumferential magnetic anisotropy formed by coupling between the negative magnetostriction and tensile stress transforms to an axial anisotropy if the sign of magnetostriction changes to positive due to stress reduction (as follows from Equation (5) and demonstrated in Figure 2). The hysteresis loop changes a shape from a flat loop to a rectangular loop. However, such stress relief is not reversible and cannot be used in temperature sensors. Moreover, practical applications demand high temperature stability.

A strong temperature dependence of the characteristic magnetic parameters is observed near the Curie temperature, which is lowered by the addition of Ni and Cr in CoFe-based amorphous alloys. The value of $T_{c}$ is also adjusted by annealing due to chemical and topological short-range reordering.

According to a classical model that assumes unrestricted spin reorientation, the magnetization varies with temperature obeying the Langeven law:(6)$$\frac{M\left(T\right)}{M_{0}}=cothx-\frac{1}{x}\;,\;\;x=\frac{3T_{c}}{T}\;\frac{M\left(T\right)}{M_{0}}.$$

In Equation (6), all the temperatures are expressed in Kelvin, and $M_{0}$ is the magnetization at T=0 K. When approaching $T_{c}$, M is proportional to ${\left(1-T/T_{c}\right)}^{\kappa }$. From Equation (6), it follows that the critical exponent κ=0.5, whereas its value in amorphous alloys is between 0.36 and 0.45. The other magnetic parameters such as anisotropy and magnetostriction also decrease near $T_{c}$ proportionally to $M^{n},\;n=2-3$ [60,61]. Then, the magnetization behavior and related effects demonstrate considerable change near the Curie temperature.

### 3.3. Magnetoimpedance (MI) Effect in Amorphous Microwires

Amorphous microwires are known to exhibit a large change in high-frequency impedance in the presence of a DC magnetic field, which is known as the magnetoimpedance (MI) effect [62,63,64]. Since the magnetic configuration in wires modifies in response to other external stimuli (such as stress and temperature), the impedance also depends on these factors. The impedance is defined by the voltage $V_{w}$ across a magnetic wire subjected to a high frequency current i. Considering the energy balance, we have:(7)$$iV_{w}=\int_{S}^{}\left(e\times h\right)ds.$$

In Equation (7), the surface integral is over the wire, while **e** and **h** are the AC electric and magnetic fields. The voltage across the wire $V_{w}$ is found from Equation (7) using the impedance boundary condition at the wire surface, which relates the axial electric field $e_{z}$ and circular magnetic field $h_{\varphi }$:(8)$$e_{z}=\varsigma_{zz}h_{\varphi }.$$

In Equation (8), $\varsigma_{zz}$ is the diagonal component of the surface impedance tensor $\hat{\varsigma }$. Considering that $\varsigma_{zz}$ is constant along the wire surface and using the relationship $h_{\varphi }=i/2\pi a$, where a is the wire radius, the voltage $V_{w}$ in a wire of length l is expressed as:(9)$$V_{w}=\frac{l}{2\pi a}\varsigma_{zz}i.$$
The calculation of $\hat{\varsigma }$ is based on the solution of the Maxwell equations for the fields **e** and **h** completed by the relationship for the dynamic magnetization $m=\hat{\chi }h$. An analytical treatment is possible considering that the magnetic susceptibility tensor $\hat{\chi }$ is of a magnetization rotational origin and is spatially independent [34,65]. Useful expressions for $\varsigma_{zz}$ valid in a wide frequency band are:
a)strong skin effect ($\frac{\delta_{m}}{a}\ll 1,\;\delta_{m}=\frac{\delta_{0}}{\sqrt{\mu_{ef}}},\;\delta_{0}=\sqrt{2/\sigma \omega \mu_{0}}$)
(10)$$\varsigma_{zz}=\frac{1-j}{\sigma \delta_{0}}\left(\sqrt{\mu_{ef}}\cos^{2}\theta +sin^{2}\theta \right),\;\;\mu_{ef}=1+\chi$$b)weak skin effect ($\frac{a}{\delta_{0}}\ll 1\;$ but can be $\frac{a}{\delta_{m}}\approx 1\;$)
(11)$$\varsigma_{zz}=\frac{k_{m}J_{0}\left(k_{m}a\right)}{\sigma J_{1}\left(k_{m}a\right)}+\frac{2}{27}{\left(\frac{a}{\delta_{0}}\right)}^{4}\frac{\mu_{3}^{2}}{\sigma a}$$
(12)$$k_{m}=\frac{1-j}{\delta_{0}}\sqrt{\mu_{1}},\;\;\mu_{1}=1+\chi \cos^{2}\theta ,\;\;\mu_{3}=1+\chi \sin\theta \cos\theta .$$

Here, σ is the DC conductivity, θ is the angle between the static magnetization and the wire axis, $\mu_{0}$ is the permeability of vacuum, χ is the susceptibility parameter composed of the components of the tensor $\hat{\chi }$ and has a meaning of the circular susceptibility in the coordinate system with the *z*-axis directed along the static magnetization, $J_{0},\;J_{1}$ are the Bessel functions of the first kind of orders 0 and 1, respectively. Expressions (10)–(12) demonstrate that the surface impedance and high frequency voltage in magnetic wires depend on the direction of the static magnetization. Certainly, the condition of an essential skin effect $a/\delta_{m}\approx 1$ is needed, so the frequency should be sufficiently high. On the other hand, the frequency is limited by the requirement that χ>1. In soft magnetic amorphous microwires (with a diameter of about 10–20 microns), the skin depth becomes about the wire radius at MHz frequencies, and the parameter χ is relatively high up to a few GHz. Therefore, the MI effect remains large in a very wide frequency band [66,67,68]. Pulse current excitation with the duration of few nanoseconds was proposed to realize MI at megahertz frequencies for applications in field sensors [69]. At GHz frequencies, the wires with a length satisfying the antenna resonance condition may behave as magnetically tunable antennas owing to the MI effect, and may be interrogated remotely by microwave irradiation [70].

## 4. Experimental Results and Discussion

### 4.1. Stress-Sensitive Magnetization Processes in Glass-Coated Amorphous Microwires of Composition Co_71_Fe_5_B_11_Si_10_Cr_3_

For many sensing applications, extremely soft magnetic properties are of interest. As discussed in Section 2, glass-coated amorphous wires of composition Co_71_Fe_5_B_11_Si_10_Cr_3_ possess near-zero magnetostriction and are of interest to realize stress-sensitive magnetic structures. Figure 4 demonstrates the transformation in the hysteresis curves in as-prepared wire in the presence of the applied tensile stress [50]. Initially, a perfect rectangular loop is seen, implying that the anisotropy easy axis is along the wire. The loop starts to incline when sufficiently high stress (> 450 MPa) is applied, overcoming the axial anisotropy. Considering the stress dependence of the saturation magnetic field, it is deduced that the magnetostriction is small but negative in the order of $-2\times 10^{-8}$, and the parameter β is about $0.9\times 10^{-10}$ MPa^−1^. In the presence of the tensile stress, the easy anisotropy direction deviates from the wire axis toward the circumference caused by the increased contribution of the magnetoelastic anisotropy (compare with Figure 2b).

A small magnitude of the magnetostriction in the as-prepared state suggests that the internal stress relaxation due to annealing may result in larger and positive $\lambda_{s}$. Then, current annealing is used to simultaneously induce a circumferential anisotropy. The intensity of the annealing current should be chosen such that the annealing temperature is sufficiently high to allow fast kinetics of structural relaxation but smaller than the Curie temperature. For the considered alloy, the optimal annealing temperature is in the range of 200–250 ℃, which corresponds to the current density of 100–120 A/mm^2^ for the metal core diameter of 25 µm. The change in the magnetostriction due to annealing may be larger than that predicted by Equation (5), accounting for stress relaxation. Other mechanisms include modifications in atomic coordinations. Thus, after annealing by the DC current of 100 A/mm^2^, the magnetostriction becomes $\lambda_{s}=5\times 10^{-8}$ and further increases after annealing with a higher current density of 120 A/mm^2^.

The change in the hysteresis loops due to current annealing is shown in Figure 5. It is clearly seen that annealing with a moderate current density of 100–120 A/mm^2^ forms a good circumferential anisotropy, since the loop becomes inclined with a small remanence/saturation ratio. Annealing with larger current densities does not change the hysteresis shape but only reduces the coercivity in comparison with that seen for as-prepared wires. This is because the temperature during annealing exceeds the Curie temperature, and the induced anisotropy does not form. If the current density (90 mA or 183 A/mm^2^ in the considered case) corresponds to temperatures close to the crystallization temperature the loop becomes inclined, but this is caused by the change in the sign of the magnetostriction due to the formation of a nanocrystalline state.

The established magnetic configuration responds sensitively to the application of the tensile stress as shown in Figure 6 and Figure 7. The combination of positive magnetostriction and tensile stress $\sigma_{ex}$ enhances the axial easy anisotropy, which is evident from a gradual increase in the remanence/saturation ratio. In the case of Figure 6 ($I_{an}=50\;\mathrm{mA}$), sufficiently high $\sigma_{ex}>500\;\mathrm{MPa}$ causes the reverse transformations, and the loop becomes inclined when the stress exceeds this value. Therefore, the anisotropy easy-axis tends toward circumference. Such stresses are sufficient to change the magnetostriction sign according to Equation (5), and $\sigma_{ex}$ coupled with a negative magnetostriction strengthens the circumferential anisotropy. For higher $I_{an}=60\;\mathrm{mA}$ (Figure 7), the magnetostriction gets larger values due to annealing, so the external stress up to very high values—more than those of the GPa—cannot change the sign of the magnetostriction. In this case, the loop becomes a perfectly rectangular shape. The observed transformation in the anisotropy easy-axis corresponds to the modeled results of Figure 3.

### 4.2. Temperature-Dependent Magnetic Properties of Amorphous Microwires with Low Tc

To realize temperature-dependent magnetic properties in the moderate range from the room temperature 50 to 80 ℃, amorphous microwires with a reduced Curie temperature are of interest. The wires of the amorphous alloy system Fe_3.9_Co_64.82_B_10.2_Si_12_Cr_9_Mo_0.08_ having $T_{c}$ values of 62 ℃ were chosen considering their rectangular hysteresis, which preserves its shape up to $T_{c}$, as shown in Figure 8 [71]. This behavior is related with relatively large and positive magnetostriction ($\lambda_{s}\approx 5\times 10^{-7})$, so some internal stress relief is not critical. A well-defined axial anisotropy exists for any temperature $T<T_{C}$, which is quite important for applications based on fast magnetization switching.

It is known that the Curie temperature of amorphous alloys can be changed by annealing within a few percent due to chemical and topological short-range restructuring. Taking the Curie temperature in *K*, the variation in absolute values can be of the order of tens of degrees. For high-$T_{C}$ alloys, this variation is not of practical interest. If $T_{C}$ is close to the room temperature, its fine control by annealing could be of huge technical importance. The change in the Curie temperature in the considered wire is demonstrated in Figure 9a [25]. The observed behavior differs from that previously reported [22]. Typically, there is a maximum $T_{C}$ value for a certain annealing temperature corresponding to the intermediate stable equilibrium. The observed minimum may be explained by antiferromagnetic coupling between Cr and Fe, Co. Anyhow, the observed variation is from 53 to 68 ℃, which is quite high in absolute values for the considered temperature range but constitutes just about 5% in relative change expressed in Kelvin.

Annealing causes structural relaxation, which may change the micromagnetic structure. In the case of wires with a rectangular hysteresis, this change is not critical and typically only results in some increase in coercivity, as shown in Figure 9b.

Other alloy systems may also have low Curie temperatures, for example, with a high content of Ni such as Fe_5_Co_27.4_B_12.26_Si_12.26_Ni_43.08_ having a $T_{c}$ value of 48 ℃. The magnetostriction of this alloy is small and negative, so the wires have a flat hysteresis. In this case, annealing not only adjusts the Curie temperature but also induces an axial anisotropy, so the hysteresis loop changes the shape and becomes rectangular. Therefore, in general, controlling $T_{c}$ by annealing has to be done with care, considering a possible modification in the magnetic structure.

### 4.3. Fast Remagnetization for Sensory Applications

Magnetic materials that have a rectangular hysteresis have the ability to generate a sharp voltage pulse when remagnetized with a low frequency magnetic field. For practical applications, they should have a low coercivity (or switching field), so the amplitude of the AC field used for remagnetization can be made small. Amorphous microwires of some FeCo alloys have a rectangular hysteresis with the coercivity smaller than 25 A/m [72]. As we demonstrated in previous sections, the hysteresis parameters at optimized magnetic, mechanical, and thermal parameters may be regulated by the external stimuli such as mechanical stress and temperature. Then, a voltage signal with a controllable amplitude and duration can be generated. The frequency spectrum of this signal contains higher harmonics of the excitation frequency, and their amplitudes are tunable by the same external action. Since high-frequency signals can be detected with excellent sensitivity using lock-in techniques, this constitutes the basis for the development of miniature and wireless sensing elements.

Figure 10 demonstrates the voltage pulse generated from remagnetizing the as-produced microwires of the composition Co_71_Fe_5_B_11_Si_10_Cr_3_ along with the amplitudes of high harmonics. The magnetic behavior of this wire is characterized by a rectangular hysteresis, as shown in Figure 4. The application of a tensile stress high enough to change the easy anisotropy direction tilts the hysteresis curves, and the harmonics amplitudes decrease with increasing $\sigma_{ex}$; moreover, higher harmonics decrease faster [73] (Figure 10b). This means that the ratio of harmonics amplitudes can be used as sensing parameter, which may not depend on a particular technical configuration.

With an increasing fundamental frequency of excitation and wire length, the harmonics’ amplitudes increase, which is important for sensor design as these dependences are not linear, as shown in Figure 11.

The microwires annealed by a DC current (100–120 A/mm^2^ in this case) have a flat hysteresis as a result of the induced circumferential anisotropy. They restore a rectangular loop after application of the tensile stress (Figure 7). In this case, the harmonics’ amplitudes sensitively increase. Figure 12 compares the behavior of the chosen harmonics amplitude (11th in this case) versus the applied stress in as-prepared and current annealed microwires. In the latter, almost a linear dependence versus $\sigma_{ex}$ is seen in the range of small values of stress, whilst the harmonic spectrum of the as-prepared wire do not show noticeable changes for such stresses. The harmonic spectrum of the wire annealed with the current density of 122 A/mm^2^, on the other hand, does not depend on higher stresses above 300 MPa. Annealing with a lower current density may induce a very similar anisotropy but smaller magnetostriction. In this case, the harmonics’ amplitudes may show a maximum and decrease with the application of higher stresses, similar to that seen in an as-prepared wire (compare with the hysteresis loops shown in Figure 6). Therefore, for particular applications, specific annealing conditions have to be chosen.

It is important that the designed stress-sensitive harmonics spectrum do not show variations with temperature. As the wires were pre-annealed by current, we can expect temperature-stable behavior. This is confirmed by examining the hysteresis loops of the annealed wires at elevated temperatures, which are given in Figure 13. Neither stress-free nor loaded wires reveal noticeable changes in the hysteresis loops for temperatures up to 80 ℃.

For amorphous microwires that preserve the rectangular hysteresis in the vicinity of the Curie temperature, the harmonics’ amplitudes decrease sharply when approaching the Curie temperature, as shown in Figure 14. The most sensitive temperature range of the spectrum variation is near the Curie temperature where the magnetization starts to drop. In the case of a Fe_3.9_Co_64.82_B_10.2_Si_12_Cr_9_Mo_0.08_ microwire having $T_{C}=$ 61.5 ℃, this range is within 55–61 ℃. Since the Curie temperature can be fine-tuned by annealing and alloy composition, this temperature range is adjustable to suit particular applications. Thus, for composite curing, higher temperatures in the range of 150–200 ℃ are needed, and for biomedical applications such as controlling the temperature at hyperthermia treatments, a narrow temperature range of 40–45 ℃ is of interest.

The temperature range where the harmonic spectrum varies with increasing temperature may be widened using a number of wires with different Curie temperatures. Figure 15 demonstrates a near linear decrease in the amplitude of the chosen harmonics with increasing temperature in the range of 40–75 ℃.

### 4.4. Magneoimpedance for Sensory Applications

As it follows from Equations (9)–(12), the high-frequency voltage $V_{w}$ or the complex-valued impedance $Z=V_{w}/i$ depend on the direction of the static magnetization and the magnetic susceptibility. In many cases, when the temperature is far from the Curie temperature, the main contribution to the impedance change comes from the change in the magnetization direction (angle θ). A magnetic field H applied along the wire changes the angle θ if the easy anisotropy is circumferential. This configuration is known to result in the most sensitive MI: Z has a minimum at H=0, and it increases sharply as the angle θ approaches zero (wire is magnetized by H along the axis). For axial anisotropy, the impedance has a maximum at H=0, and it decreases slowly with increasing H as there is no change in the magnetization direction, so the impedance follows a week dependence of χ(H). Therefore, we can expect the transformation in the Z(H) plot if the type of the easy anisotropy changes due to the application of a mechanical stress.

Figure 16 demonstrates how the impedance plots for an as-prepared Co_71_Fe_5_B_11_Si_10_Cr_3_ wire with an axial anisotropy transform when the wire is subjected to the tensile stress [28]. This transformation is consistent with the change in the hysteresis loops (compare with Figure 4).

Initially, the impedance has a maximum at H=0 and with increasing $\sigma_{ex}$ values, it splits into two peaks, which reflects the change in the easy anisotropy direction (from axial to circumferential) as the magnetostriction becomes negative. Then, for small fields, the impedance monotonically decreases with increasing $\sigma_{ex}$, although the sensitive change in Z occurs for higher stresses, as shown in Figure 17. When measuring the impedance, the load was applied in the middle of the wire, and the imposed stress is nonuniform along the length. On the other hand, a uniform stress was applied to the wire during the hysteresis loop measurements. This difference explains some quantitative disagreement, but the trend in the impedance and hysteresis loop transformations under the effect of tensile stress is consistent. The largest variation in impedance occurs when no external field is applied: the relative change is nearly 100% in the range of 200–600 MPa. The application of the field increases the insensitive range with respect to $\sigma_{ex}$, since the axial field helps to hold the magnetization along the axis.

The impedance of current annealed wires versus stress behaves in an opposite manner [16] (see Figure 18). Owing to the induced circumferential anisotropy, the impedance has a minimum at zero field and two symmetrical maxima when the field is about the anisotropy field ($H=\pm H_{k})$. It is seen that the values of $H_{k}$ found from the impedance plots are considerably higher than its values estimated from the hysteresis curves (which can be taken as the saturation field). In particular, a large difference is noticed for the wire annealed with a higher current of 60 mA (122 A/mm^2^). Since the impedance characteristics at MHz frequencies are determined by the properties of the surface layer, this observation implies that the induced anisotropy is strongly nonuniform and is much larger at the surface.

The application of $\sigma_{ex}$ forces the impedance maxima to merge since the current annealing also changes the magnetostriction, which becomes large and positive. Then, the application of a tensile stress contributes to the axial anisotropy; therefore, the impedance versus field plots tend to have a single peak. In the case of a moderate annealing current, higher stresses may still change the magnetostriction sign, and the impedance behavior changes again: the two peaks in Figure 18a are seen at larger $H_{k}$ values.

A strong change in impedance can be expected near the Curie temperature $T_{c}$ where both the magnetization and effective anisotropy decrease. Owing to the decrease in M and $H_{K}$, the ferromagnetic resonance frequency $f_{res}=\gamma \sqrt{H_{K}\left(H_{K}+M\right)}/2$ (γ is the gyromagnetic constant and $H_{K}$ is the effective anisotropy field) decreases near $T_{c}$ and the dispersion region of $\hat{\chi }$ shifts to lower frequencies; therefore, the high-frequency properties are suppressed [69]. However, at low frequencies, the initial susceptibility $\chi_{in}\approx M/H_{K}$ increases when a temperature is approaching $T_{c}$, since the effective anisotropy $\tilde{K}\propto M^{n}$ and n>2. Then, at lower frequencies, the impedance may initially increase near $T_{c}$ but eventually will drop as the resonance frequency tends to zero. A non-monotonic behavior of the impedance when approaching $T_{c}$ was observed in [71] for frequencies lower than 100 MHz. For higher frequencies, the susceptibility decreases with temperature following the decrease in $f_{res}$, and the impedance experiences a monotonic decrease, as demonstrated in Figure 19. It is seen that there is a very weak dependence on the magnetic field, since the magnetic susceptibility at high frequencies depends weakly on changing the field. Figure 20 shows the impedance versus temperature plots.

## 5. Conclusions

We investigated the performance of ferromagnetic amorphous microwires as mechanical stress and temperature sensors utilizing the effects of fast remagnetization and magnetoimpedance. The wires of composition Co_71_Fe_5_B_11_Si_10_Cr_3_ in as-prepared and current-annealed states are used to demonstrate a strong influence of the tensile stress on the magnetization process, as well as the generation of a high harmonics spectrum and magnetoimpedance. The stress effect is enhanced after current annealing, owing to a competition between the induced and magnetoelastic anisotropies. The wires with a low Curie temperature, for example, of composition Co_23.67_Fe_7.14_Ni_43.08_B_13.85_Si_12.26_ ($T_{c}=$ 61–62 ℃) which can be fine-tuned by annealing, demonstrate a rectangular hysteresis even for temperatures very close to the Curie temperature. The existence of well-defined axial easy anisotropy up to the Curie temperature is important to realize temperature-sensitive fast remagnetization and magnetoimpedance. At high frequencies beyond the frequency of the ferromagnetic resonance, the impedance monotonically decreases when approaching $T_{c}$: at a frequency of 300 MHz, the impedance decreases almost twice in a narrow temperature range of 50–60 ℃.

## Figures and Tables

**Figure 1 sensors-19-05089-f001:**
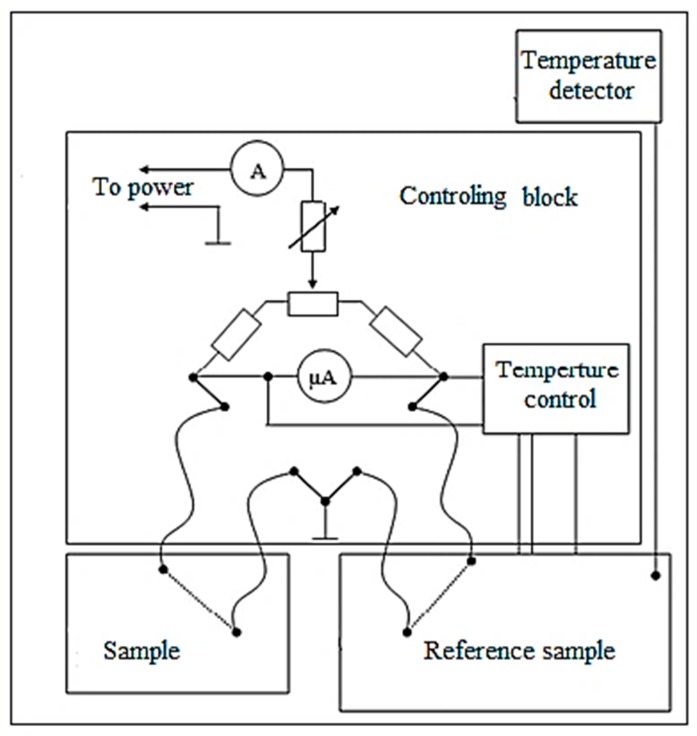
Schematic diagram of a setup for current annealing with remote temperature control.

**Figure 2 sensors-19-05089-f002:**
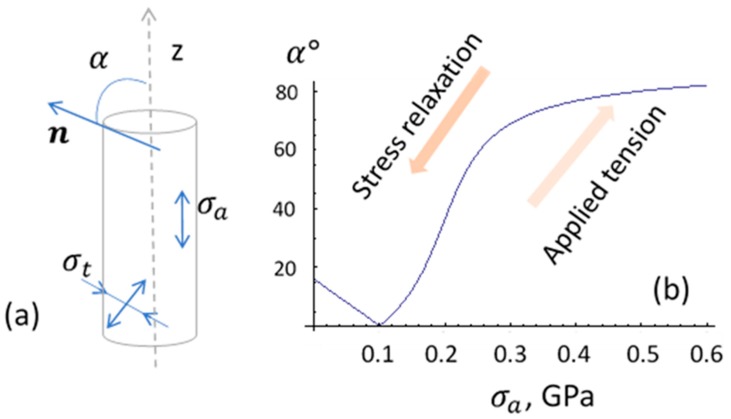
In (**a**), schematic representation of various stress contributions to the easy anisotropy of amorphous wire; in (**b**), dependence of the angle α between the easy axis n and the wire axis z on tensile stress $\sigma_{a}$. Parameters used for calculation: $\lambda_{s0}=2.5\times 10^{-8}$, $\beta =0.9\times 10^{-10}{\mathrm{MPa}}^{-1}$, and $\frac{2K}{M_{s}}=160\;A/m$. In order to describe a smooth transformation in the easy angle α, a small torsion stress $\sigma_{t}=40\;\mathrm{MPa}$ was introduced.

**Figure 3 sensors-19-05089-f003:**
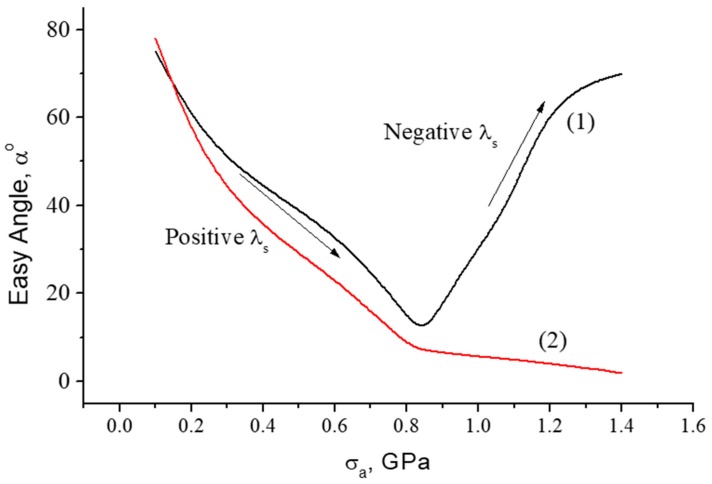
Dependence of the angle α between the easy axis n and the wire axis z on tensile stress $\sigma_{a}$ in wires with induced circular anisotropy and positive magnetostriction. Parameters used for calculation: $\beta =1.2\cdot 10^{-10}\;{\mathrm{MPa}}^{-1}$, $\frac{2K_{u}}{M_{s}}=180\;A/m$. In order to describe a smooth transformation in the easy angle α, a small torsion stress $\sigma_{t}=40\;\mathrm{MPa}$ was introduced. For Curve (1) $-\lambda_{s0}=5\times 10^{-8}$; for Curve (2) −$\lambda_{s0}=2\times 10^{-7}$.

**Figure 4 sensors-19-05089-f004:**
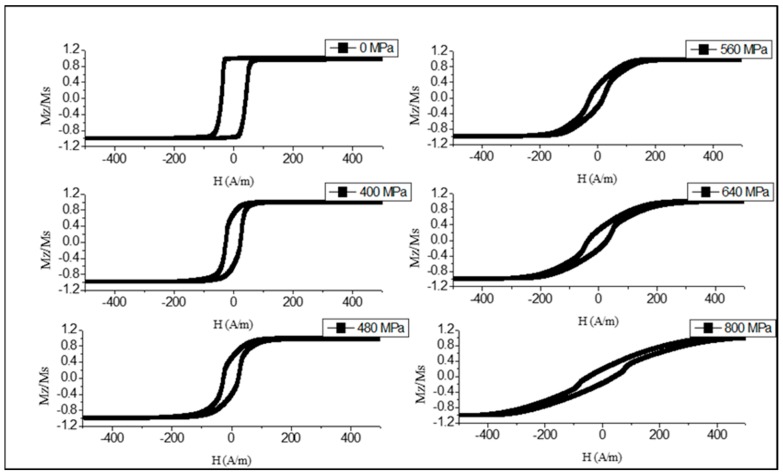
Effect of tensile stress on hysteresis loops of glass-coated amorphous Co_71_Fe_5_B_11_Si_10_Cr_3._ microwires in the as-prepared state.

**Figure 5 sensors-19-05089-f005:**
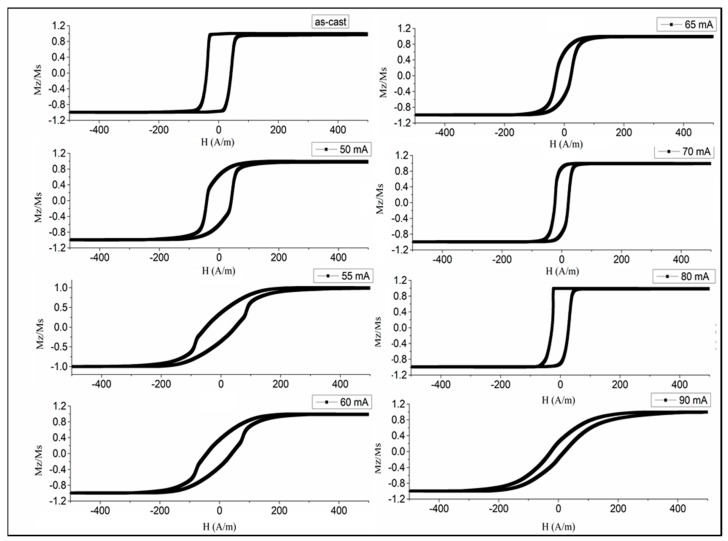
Effect of current intensity during annealing on the hysteresis loops of glass-coated amorphous Co_71_Fe_5_B_11_Si_10_Cr_3_ microwires (50 mA corresponds to a current density of 102 A/mm^2^ for the wire diameter of 25 microns).

**Figure 6 sensors-19-05089-f006:**
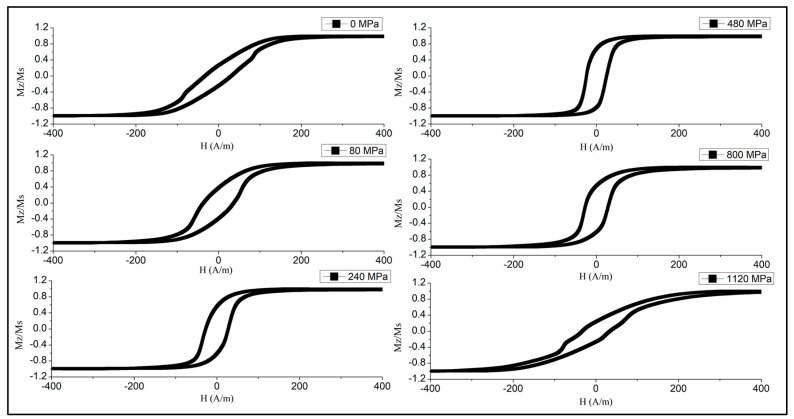
Effect of tensile stress on the hysteresis loops of glass-coated amorphous Co_71_Fe_5_B_11_Si_10_Cr_3_ microwires after current annealing at 50 mA (corresponds to the current density of 102 A/mm^2^) for 60 min.

**Figure 7 sensors-19-05089-f007:**
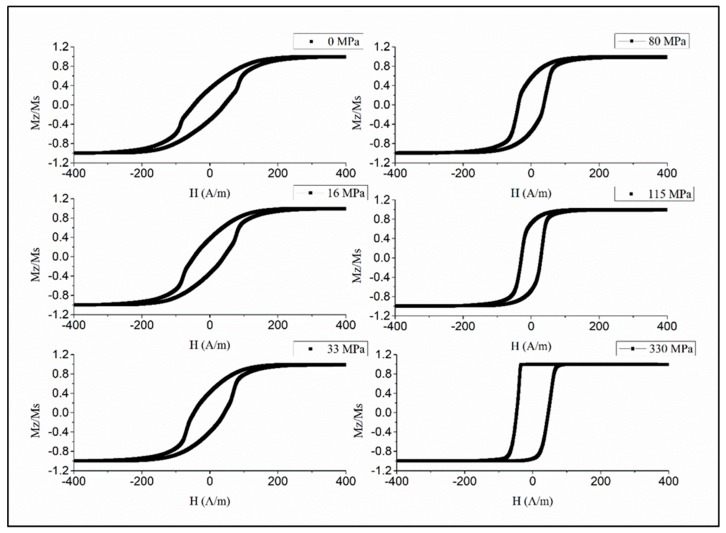
Effect of tensile stress on the hysteresis loops of glass-coated amorphous Co_71_Fe_5_B_11_Si_10_Cr_3_ microwires after current annealing (at 60 mA (122 A/mm^2^) for 60 min).

**Figure 8 sensors-19-05089-f008:**
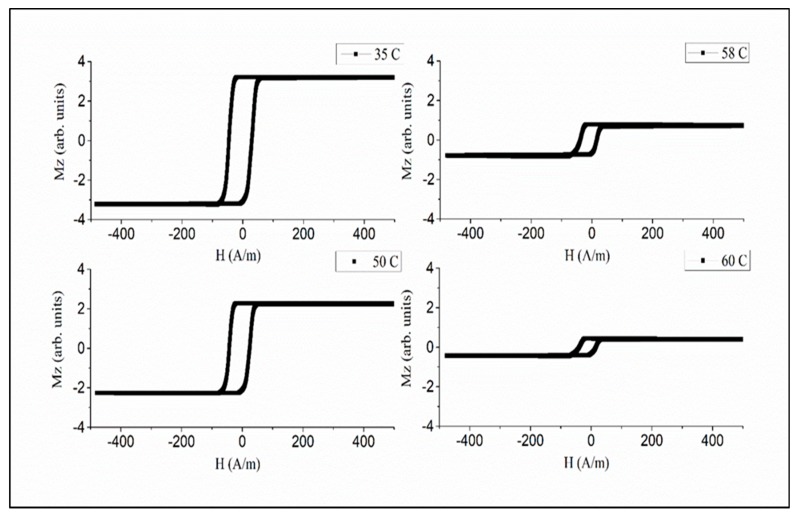
Magnetization loops of glass-coated Fe_3.9_Co_64.82_B_10.2_Si_12_Cr_9_Mo_0.08_ microwires for different temperatures, including near $T_{C}=$ 62 ℃ obtained by the induction method. The loops are normalized by the saturation value at room temperature. The wire had the total diameter of 27.3 µm and the metallic core diameter of 17.7 µm.

**Figure 9 sensors-19-05089-f009:**
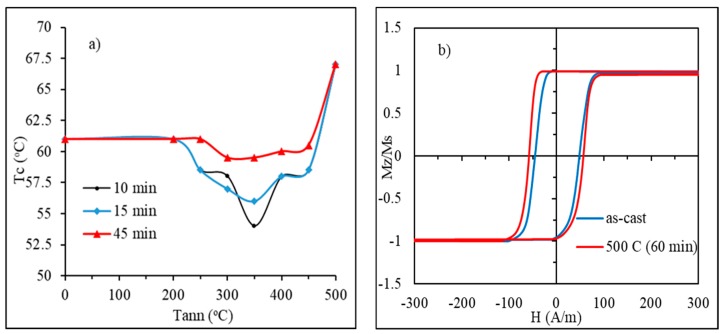
Effect of annealing on the Curie temperature and hysteresis loops for amorphous Fe_3.9_Co_64.82_B_10.2_Si_12_Cr_9_Mo_0.08_ microwire (in as-prepared state $\;T_{c}=$ 61.5 ℃). In (**a**), $T_{c}$ vs. annealing temperature for different annealing times; in (**b**), the hysteresis curves in the as-prepared state and after annealing with a high annealing temperature of 500 ℃.

**Figure 10 sensors-19-05089-f010:**
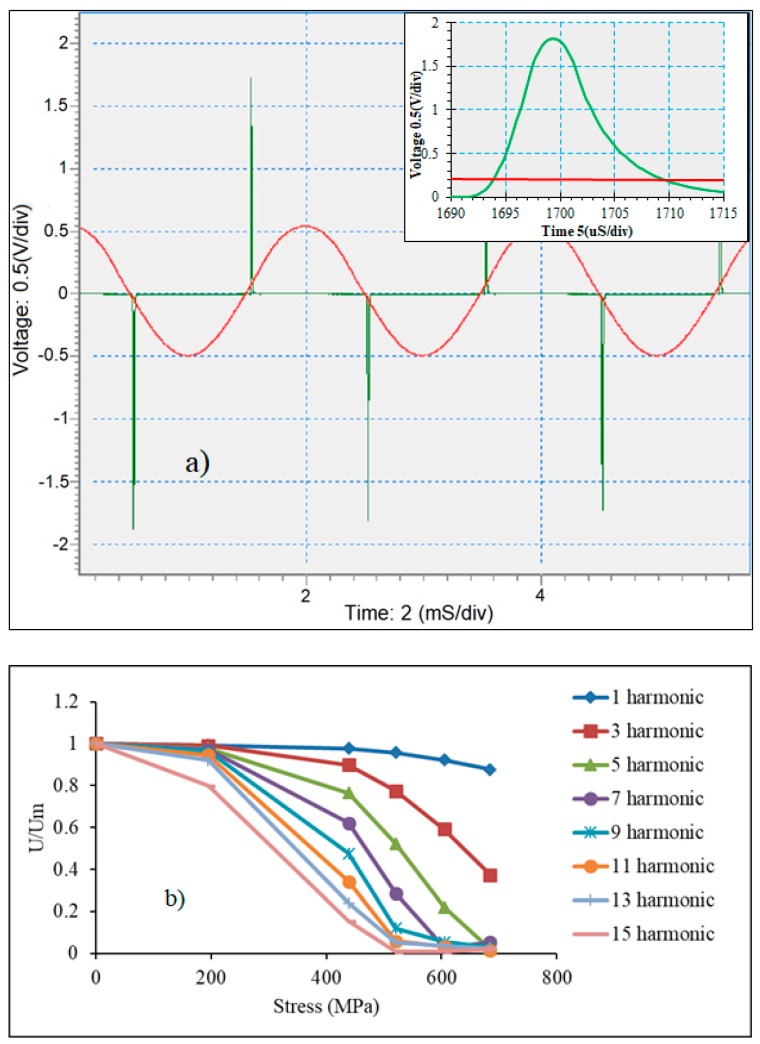
Voltage pulse at remagnetization (**a**) and amplitudes of high harmonics vs. $\sigma_{ex}$ (**b**) for glass-coated amorphous Co_71_Fe_5_B_11_Si_10_Cr_3_ microwires in the as-prepared state.

**Figure 11 sensors-19-05089-f011:**
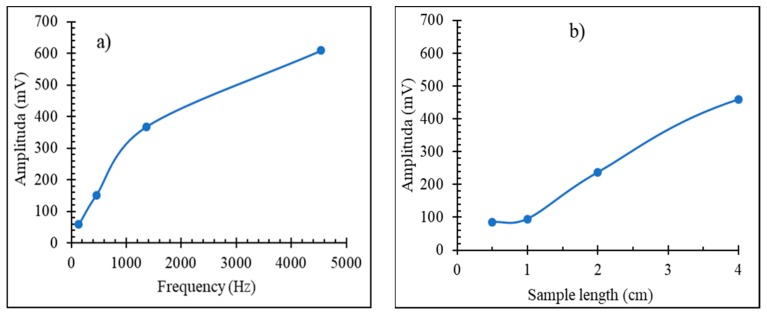
Effect of fundamental frequency of the magnetizing field in (**a**) and sample length in (**b**) on the amplitude of high harmonics (11th harmonic) of glass-coated amorphous Co_71_Fe_5_B_11_Si_10_Cr_3_ microwires in the as-prepared state.

**Figure 12 sensors-19-05089-f012:**
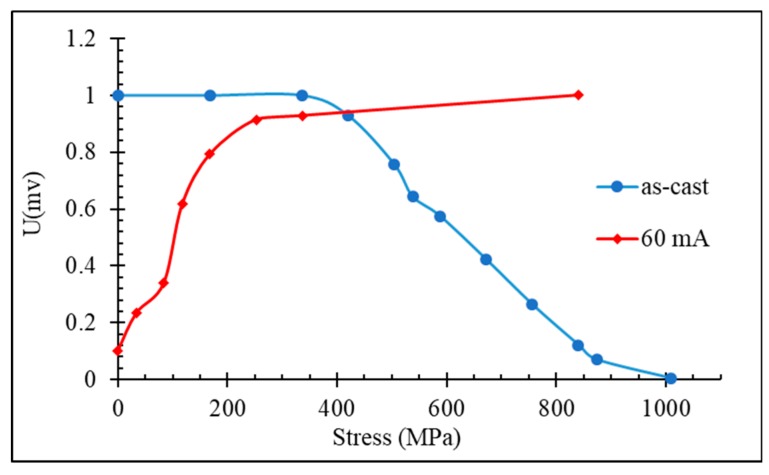
Effect of tensile stress on the amplitude of 11th harmonics of glass-coated amorphous Co_71_Fe_5_B_11_Si_10_Cr_3_ microwires in an as-prepared state and after current annealing at 60 mA (122 A/mm^2^) for 60 min (compare with the change in the hysteresis curves shown in Figure 7).

**Figure 13 sensors-19-05089-f013:**
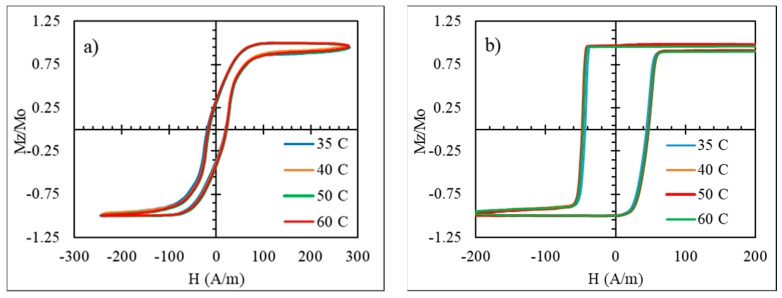
Effect of temperature on hysteresis loops of glass-coated amorphous Co_71_Fe_5_B_11_Si_10_Cr_3_ microwires after current annealing (at 60 mA (122 A/mm^2^) for 60 min). In (**a**), without stress; in (**b**), with stress of 480 MPa (the stress was constant while the temperature was changing).

**Figure 14 sensors-19-05089-f014:**
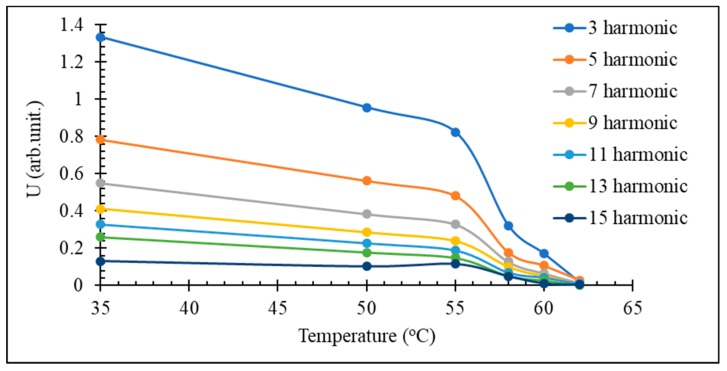
Harmonic spectrum vs. temperature of glass-coated amorphous Fe_3.9_Co_64.82_B_10.2_Si_12_Cr_9_Mo_0.08_ microwire having $T_{C}=$ 61.5 ℃ in an as-prepared state.

**Figure 15 sensors-19-05089-f015:**
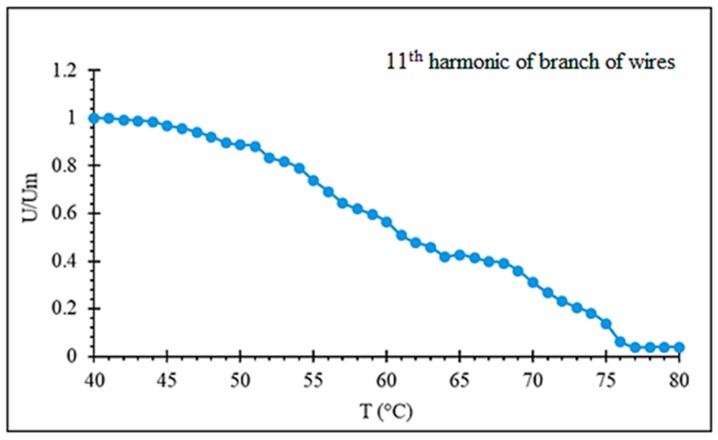
Effect of temperature on amplitude of 11th harmonics of a branch of five wires with different $T_{c}$ (1: Fe_3.9_Co_64.82_B_10.2_Si_12_Cr_9_Mo_0.08_ as-cast with $T_{C}=$ 61.5 ℃, 2: same composition, annealed at 350 ℃ for 5 min with $T_{C}=$ 55 ℃; 3: same composition, annealed at 500 ℃ for 5 min with $T_{C}=$ 66 ℃; 4: Fe_4.9_Co_64.82_B_10.2_Si_12_Cr_8_Mo_0.08_ as cast with $T_{C}=$ 69 ℃; and annealed at 500 ℃ for 25 min with $T_{C}=$ 78 ℃).

**Figure 16 sensors-19-05089-f016:**
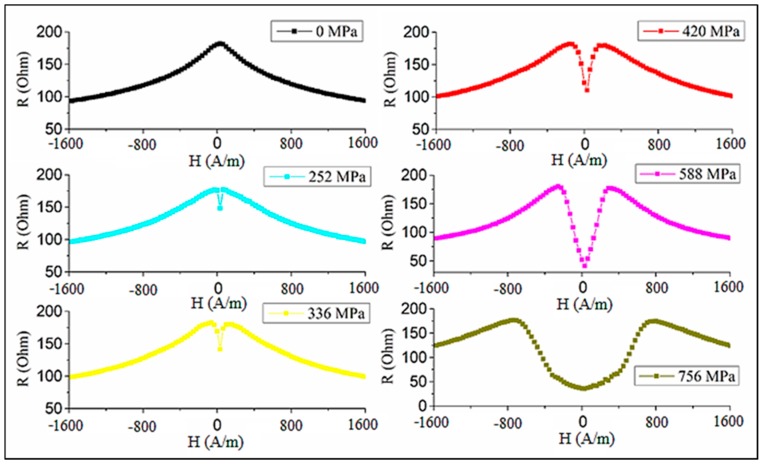
Effect of tensile stress on magnetoimpedance in as-prepared amorphous Co_71_Fe_5_B_11_Si_10_Cr_3_ microwires. Real part of the impedance R(H) is given. Frequency is 50 MHz.

**Figure 17 sensors-19-05089-f017:**
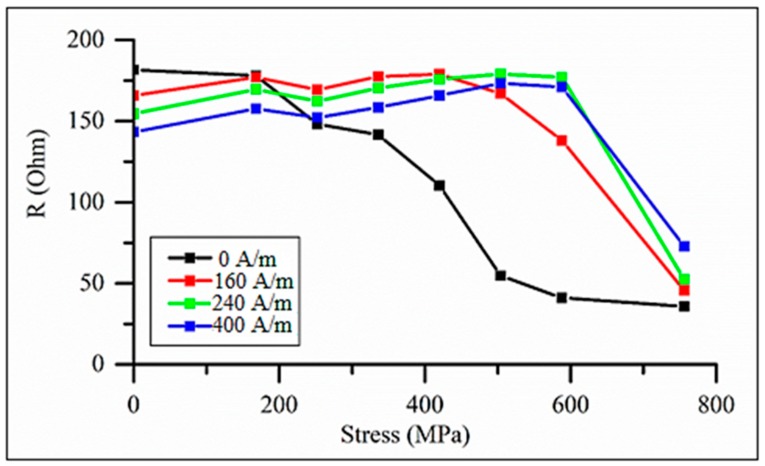
Real part of impedance vs. tensile stress for as-prepared amorphous Co_71_Fe_5_B_11_Si_10_Cr_3_ microwires with an external field as a parameter. Frequency is 50 MHz.

**Figure 18 sensors-19-05089-f018:**
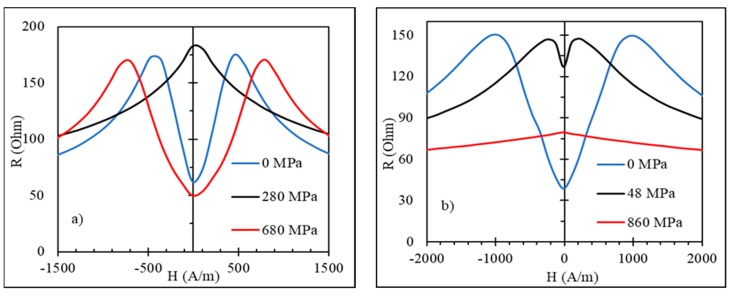
Effect of tensile stress on magnetoimpedance (MI) of glass-coated amorphous Co_71_Fe_5_B_11_Si_10_Cr_3_ microwires after current annealing: (**a**) at 50 mA (102 A/mm^2^) for 60 min and (**b**) at 60 mA (122 A/mm^2^) for 60 min. Frequency is 50 MHz.

**Figure 19 sensors-19-05089-f019:**
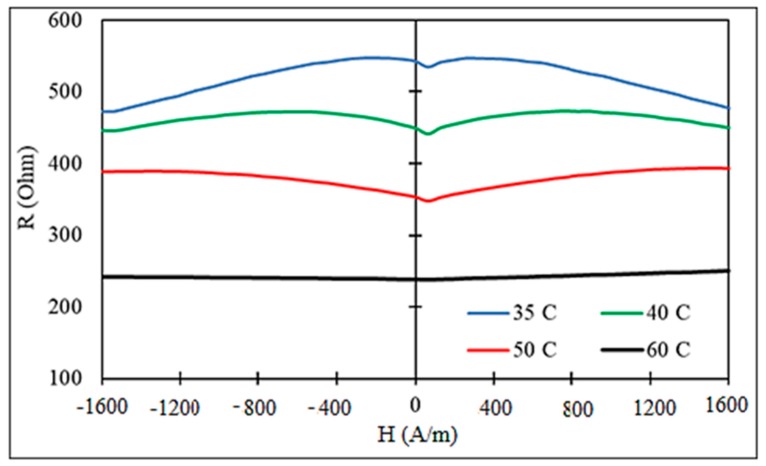
Real part of the impedance of glass-coated microwires of the composition Fe_3.9_Co_64.82_B_10.2_Si_12_Cr_9_Mo_0.08_ having $T_{c}=$ 61 ℃ for various temperatures. Frequency is 300 MHz.

**Figure 20 sensors-19-05089-f020:**
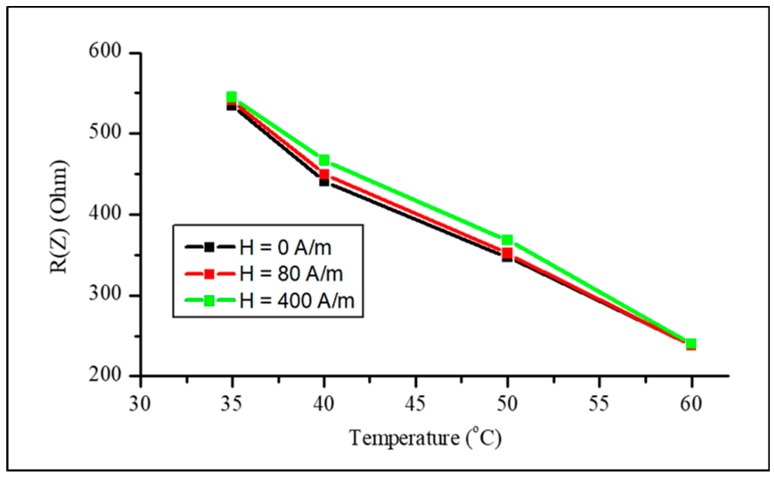
Real part of impedance (at 300 MHz) vs. temperature for as-prepared amorphous Fe_3.9_Co_64.82_B_10.2_Si_12_Cr_9_Mo_0.08_ microwires for fixed external magnetic fields.

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
