# Peer review of "Soft Magnetic Amorphous Microwires for Stress and Temperature Sensory Applications"

_sensors, 2019, doi:10.3390/s19235089_

Round 1

Reviewer 1 Report

This paper presents an extensive and detailed study, but there is almost no mention of experimental methods. How was stress applied? Was the generated stress uniform across the samples? These are important questions that need to be answered first before the paper can be further reviewed and scrutinized. Important experimental details are missing at this stage and should be provided in a revision.

Reviewer 2 Report

The authors have produced a review of stress and temperature dependence of the magnetic properties of glass-coated amorphous microwires. They consider two alloy compositions. It would be helpful to broaden the context of this paper by citing specific device applications of the properties considered.

Then the authors could consider other compositions which might exhibit the same properties which they have studied. This would mean citing other groups which have worked with glass-coated microwires as their work relates to the problems being considered here. If this is a review article then it must have a broader scope and cannot focus only on the work of this group. 

The theory describing the effects is spread out in the results and discussion section. Please bring it all together and place it before results and discussion.

Round 2

Reviewer 1 Report

The authors may not have understood what I was driving at. If they generated stress by hanging a load in the middle of the wire, the stress will be very non-uniform along the length of the wire. The paper however assumes uniform stress. The authors need to acknowledge that the assumption of uniform stress is inexact and therefore results can only be viewed as approximate as opposed to accurate.
